

# Multi-year, spatially extensive, watershed scale synoptic stream chemistry and water quality conditions for six permafrost-underlain Arctic watersheds

Arial J. Shogren[1], Jay P. Zarnetske[1], Benjamin W. Abbott[2], Samuel Bratsman[2], Brian Brown[2], Michael
P. Carey[3], Randy Fulweber[4], Heather E. Greaves[4], Emma Haines[1], Frances Iannucci[4, 5], Joshua C.
Koch[3], Alexander Medvedeff[5], Jonathan A. O'Donnell[6], Leika Patch[2], Brett A. Poulin[7,8], Tanner J.
Williamson[1], William B. Bowden[5]

[1] Earth & Environmental Sciences Department, Michigan State University, East Lansing Michigan 48824, USA
[2] Plant & Wildlife Sciences Department, Bringham Young University, Provo, Utah, 84602, USA
[3] Alaska Science Center, U.S. Geological Survey, Anchorage, Alaska, 99508, USA
[4] Institute of Arctic Biology, University of Alaska Fairbanks, Fairbanks, Alaska, 99775, USA
[5] Rubenstein School of Environment and Natural Resources, University of Vermont, Burlington, Vermont 05405, USA
[6] Arctic Network, National Park Service, Anchorage, Alaska, 99501, USA
[7] Water Mission Area, U.S. Geological Survey, Boulder, Colorado, 80303, USA
8 Department of Environmental Toxicology, University of California Davis, Davis, California, 95616, USA

*Correspondence to*: Arial J. Shogren (shogrena@msu.edu)

**Abstract.** Repeated sampling of spatially distributed river chemistry can be used to assess the location, scale, and stability of carbon and nutrient contributions to watershed-scale exports. Here, we provide a comprehensive set of water chemistry measurements and secondary ecosystem metrics describing the biogeochemical conditions of permafrost-affected Arctic watershed networks. These data were collected in watershed-wide repeated synoptic campaigns across six rivers across northern Alaska. Three watersheds are associated with the Arctic Long-Term Ecological Research (ARC LTER) site at Toolik Field Station (TFS), which were sampled seasonally each June and August from 2016 to 2018. Three watersheds were associated with the National Park Service (NPS) of Alaska and the US. Geological Survey (USGS), and were sampled annually from 2015 to 2019. Extensive water chemistry characterization included carbon species, dissolved nutrients, and anions and cations. The objective of the sampling designs and data acquisition was to generate a dataset to support the estimation of ecosystem metrics that describe the dominant location, scale, and overall stability of ecosystem processes in the Arctic. These metrics are: (1) subcatchment leverage, (2) variance collapse, and (3) spatial stability. Both water chemistry concentrations and secondary metrics are available at the National Park Service Integrated Resource Management Application portal (https://doi.org/10.5066/P9SBK2DZ) and within the Environmental Data Initiative LTER Data Portal (https://doi.org/10.6073/pasta/258a44fb9055163dd4dd4371b9dce945).



**Plaintext Summary.** Sampling various points in an entire river network over a short period of time (~a few hours) provides a synoptic "snapshot" in time of the water chemistry in a watershed. Here, we describe two unique datasets which captured

river chemistry snapshots in six permafrost-impacted watersheds in northern Alaska. We present how these repeated snapshots can be used to inform predictions for carbon, nutrient, and other solutes in landscapes that are rapidly changing as a direct result of climate change.

## 1 Introduction

        Watershed chemistry and water quality studies frequently involve a trade-off between sampling extent (i.e., how

much heterogeneity the study captures) and spatial scale (i.e., how much spatial extent is covered) in river networks (Abbott et al., 2018; Burns et al., 2019; Ward et al., 2019). This trade-off is especially apparent in remote settings, such as the Arctic, where logistical constraints and high operational costs often force researchers to choose among these sampling approaches. Initial assessments are typically performed at the plot (terrestrial studies, <1–100 m$^2$) (Keller et al., 2007; Prager et al., 2017) or reach-scale (stream studies, 100-1000 m) (Kling et al., 2000; Docherty et al., 2018). While these intensive studies often

allow greater understanding of the underlying processes controlling solute transport and transformations, up-scaling observations from small-scale studies is extremely challenging (Wiens, 1989; Thrush et al., 1997). Often, processes that occur at the plot- or reach-scales may not be constant over time (Kareiva and Andersen, 1988). In addition, experiments at smaller scales cannot fully capture the extent of heterogeneity of the full watershed network, nor do they always reveal emergent patterns and processes (Sivapalan, 2003; McDonnell et al., 2007). Here, we provide a rich and rare watershed

chemistry dataset that contains spatially distributed hydrological, ecological, and geochemical properties measured across entire watershed scales (<1 to >1000 km$^2$) spanning multiple watersheds and years. This dataset is unique and may facilitate multiple disciplinary as well as interdisciplinary studies of Arctic systems.

        Unlike the watershed-scale dataset presented here, most chemistry and water quality assessments conducted in the Arctic and elsewhere are typically done via measurements of water flow and chemistry at river outlets (McClelland et al.,

2006, 2007; Tank et al., 2016; Toohey et al., 2016; Shogren et al., 2020). The flow of water integrates biogeochemical signals, such that river chemistry at the watershed outlet, contains information about both terrestrial and aquatic biogeochemical processes that occurred upstream in the network (Temnerud et al., 2010; Vonk et al., 2019). Indeed, using sampling and monitoring approaches that capture the watershed outlet response over time has logistic and safety advantages



for site access. Further, the recent application of novel sensor technology has enabled high-frequency watershed-scale

studies (Shogren et al., 2021). For example, the paired high-frequency flow and a limited set of chemical properties for the

watersheds in this data paper are available at the Arctic Data Center (Zarnetske et al., 2020b, c, a). While these riverine

measurements are inherently valuable from remote regions (Laudon et al., 2017; Shogren et al., 2021), there are still

challenges related to using watershed-scale measurements to diagnose primary drivers of solute export (Burns et al., 2019).

Large-scale measurements are the result of variable inputs which are "buffered" as the signals are mixed and propagated

over the large watershed network (Creed et al., 2015). Indeed, both reach- and watershed-scale frameworks present

important opportunities to constrain the uncertainty of biogeochemical fluxes from Arctic ecosystems (Kicklighter et al.,

2013). However, watershed outlet observations are often difficult to directly link back to intermediate-scale processes or

specific watershed locations that drive solute transport from terrestrial to aquatic ecosystems (Hoffman et al., 2013; Collier

et al., 2018).

70       Spatially extensive or "synoptic" sampling frameworks, such as contained in this data paper, have the advantage of

providing information about the distribution of signals across the entire watershed network, and can be used to complement

watershed outlet monitoring. With a synoptic sampling design, researchers can capture the spatial extent of nested

subcatchments and therefore assess multiple scales of processes driving watershed chemistry as signals are propagated down

the network (Abbott et al., 2018; Shogren et al., 2019). Though synoptic campaigns are logistically challenging (Yi et al.,

2010), the informative "snapshot" of the changes along stream networks at a given point in time allows empirical assessment

of biogeochemical signals at intermediate spatial scales (Abbott et al., 2018; Shogren et al., 2019). In recent years, synoptic

campaigns have focused on solute distribution in temperate river systems (Gardner and McGlynn, 2009; Byrne et al., 2017;

Abbott et al., 2018; Dupas et al., 2019). While spatially explicit campaigns in permafrost-underlain streams have largely

lagged behind those in temperate rivers (Kling et al., 2000; Bowden, 2013; Shogren et al., 2020), their application presents

an opportunity to characterize the fate of carbon and nutrients in a rapidly changing Arctic. Therefore, measuring the spatial

distribution of water chemistry in high latitude river networks has significant scientific value to both Arctic systems and

global biogeochemical and climatic conditions (Bring et al., 2016; Wrona et al., 2016).

The datasets presented here were derived from repeated synoptic samplings in six Arctic watersheds in northern Alaska, which represent several distinct high latitude landscapes (tundra, boreal, alpine, Figure 1). Within this manuscript,

we illustrate the utility of such data via a set of initial watershed chemistry analyses on a suite of ecologically significant reactive solutes including dissolved organic carbon (DOC), nitrogen (e.g., nitrate, $N-NO_3^-$; ammonium, $N-NH_4^+$; dissolved organic nitrogen, DON; total dissolved nitrogen, TDN), phosphorous (soluble reactive phosphorus, SRP; total dissolved phosphorus, TDP), as well as a suite of geochemically significant anions and cations (e.g., calcium, $Ca^{2+}$; total iron, Fe; dissolved silica, DSi; *see Table 1 for full list of analytes*). In addition, we use these datasets to introduce simple metrics for

biogeochemical solutes: *variance collapse*, *subcatchment leverage*, and *spatial stability* (Abbott et al., 2018; Shogren et al., 2019). These new metrics help illustrate more nuanced assessments of watershed signals that become possible with watershed-scale, spatially extensive synoptic data. That is, they capture what spatial scale is the most relevant in explaining spatial variation in solute concentration, how much "influence" each sampling site has on the watershed budget, and whether or not samples from a single location are representative over time. When used in combination with metrics such as

subcatchment leverage, variance collapse, and spatial stability, synoptic sampling frameworks provide robust information on the spatial scale and configuration of major processes that contribute to biogeochemical fluxes. Ultimately, the information gleaned from these metrics is desired by a range of disciplines from ecologists to natural resource managers.

First, we use subcatchment leverage to identify nested areas within the network that exert a disproportionate influence on flux at the watershed outflow. Subcatchment leverage can be interpreted as the contribution of the subcatchment

to watershed mass flux where the value can be negative (indicating a net source of solute production), positive (indicating a net sink for solute removal), or near-zero (reflecting net conservative solute behaviour). Estimating leverage allows identification of specific subcatchments with disproportionate influence on material export, defined here as high leverage. Subcatchments with high leverage behave as a strong source or sink within the watershed network, strongly influencing the resulting concentrations at the outflow, and can be selected as sites for further mechanistic study or monitoring. Likewise,

the direction and magnitude of leverage averaged across the entire watershed contains information about net solute removal and production. Second, we examine how patch size controls solute production and removal by identifying thresholds of concentration variance collapse. We generally expect the amplitude of solute variability to decrease moving downstream

from headwaters to larger systems (Creed et al., 2015). Higher solute concentration variability within the watershed is most often observed in headwaters (Wolock et al., 1997; Temnerud and Bishop, 2005), whereas downstream reaches are less

likely to have extremely high or low concentrations because they integrate multiple upstream source or sink processes (Abbott et al., 2018). Therefore, the size of nutrient sources and sinks in the landscape can be assessed by the spatial scale of the variance collapse of concentration among watershed reaches (Abbott et al., 2018; Shogren et al., 2019). The threshold of variance collapse is similar to the elementary representative area concept (Zimmer et al., 2013, p.20), where the threshold represents the spatial scale at which landscape "patches" or processes throughout the watershed network that produce and

remove solutes are effectively integrated. Lastly, the spatial stability metric can be used to assess whether a given site is representative (i.e., stable), or if patches restructure in space between sampling campaigns (i.e., unstable). Spatial stability effectively quantifies the temporal representativeness of an instantaneous measurement at a given site, informing future watershed study design and data analysis of extant data (Kling et al., 2000; Shogren et al., 2019). Taken together, watershed metrics developed from the extensive watershed scale synoptic data presented in this paper provide new insights into

ecological and geochemical processes connecting land to water in Arctic watersheds.

## 2 Study Location & Design

### 2.1 Study Watersheds

#### 2.1.1 Arctic LTER sites at Toolik Field Station

The Arctic Long-Term Ecological Research (ARC LTER) site based out of Toolik Field Station (TFS) is located in the

foothills of the Brooks Range on the North Slope of Alaska, USA (mean elevation 720 m). We conducted surveys in three watersheds near TFS: the Kuparuk River, Oksrukuyik Creek, and Trevor Creek. The three study watersheds were chosen as they spanned dominant circumarctic vegetation types, permafrost characteristics, and hydrologic conditions (Table 1). Further, the climate, morphology, and ecology of the sites and region have been previously described (Hobbie and Kling, 2014).

• The **Kuparuk River** (68.64816, -149.41152, Figure 2A) is a meandering stream flowing through primarily tundra vegetation, located about 10 km northeast of TFS. The Kuparuk River includes a long-term monitoring site for the ARC LTER, used as a site for ecological study and monitoring since 1979. From 1983-2016, the 4[th]



order reach of the Kuparuk River was used for a whole-stream fertilization study (Peterson et al., 1993; Slavik et al., 2004; Iannucci et al., 2021), where phosphorous ($H_3PO_4$) was continuously added to assess response to nutrient fertilization. As the Kuparuk River continues north, it meets a large aufeis (ice) field (Yoshikawa et al., 2007; Terry et al., 2020).

- **Oksrukuyik Creek** (68.68740, -149.095, Figure 2B) is a clear-water, low-gradient stream meandering through primarily tundra landscape, with intermittent presence of stream-lake connectivity (Shogren et al., 2019). Oksrukuyik Creek is also an ARC LTER long-term monitoring site, approximately 20 km northeast of TFS.

- **Trevor Creek** (68.28482, -149.350063, Figure 2C) is a mountainous alpine stream, draining into the Atigun River watershed, located 30-km south of TFS. Trevor Creek drains primarily steep, rocky slopes with limited heath and willow vegetation. The majority of stream runoff is generated by precipitation and snowmelt.

As a result of long-term study and a sustained commitment to data stewardship, the ARC LTER and TFS hosts an extensive catalogue of terrestrial, aquatic, and atmospheric data that are complementary to the data presented in this publication. For more information, please see the LTER data catalogue (https://arc-lter.ecosystems.mbl.edu/data-catalog), in addition to the abiotic and biotic monitoring data from the TFS Spatial and Environmental Data Center (https://toolik.alaska.edu/edc/index.php).

### 2.1.2 National Parks Service and U.S. Geological Survey Sites

We also sampled three watersheds associated with the National Park Service (NPS) Arctic Inventory and Monitoring Network and a project funded by the U.S. Geological Survey's (USGS) Changing Arctic Ecosystem program. The Agashashok and Cutler River watersheds are within Noatak National Preserve and the Akillik River watershed is within Kobuk Valley National Park. All three watersheds are situated near the northern extent of Alaska's boreal forest, where tree line is expanding (Suarez et al., 1999), and subcatchments vary in areal extent of forested versus tundra land cover. The study sites vary with respect to permafrost characteristics, including soil texture, ground ice content, and subsurface hydrology (O'Donnell et al., 2016). Evidence suggests stream chemistry varies across these watersheds, including the form, amount, and age of dissolved carbon (O'Donnell et al., 2020).



- The **Cutler River** (67.845, -158.316, Figure 3A) flows north out of the Baird Mountains through gently rolling tundra into the upper Noatak River. The watershed is underlain by ice-rich glaciolacustrine deposits (O'Donnell et al., 2016), and soils tend to be organic-rich and poorly drained. Vegetation is dominated by moist acidic tundra and wet sedge meadows.

- The **Akillik River** (67.201, -158.572, Figure 3B) flows south out of the Baird Mountains and into the Kobuk River downstream of the village of Ambler, Alaska. The river passes through alpine terrain in the headwaters before draining terrain comprised of ice-rich loess in the lower reaches. Vegetation is a mixture of boreal spruce forests and tundra.

- The **Agashashok River** (67.268, -162.636, Figure 3C) is a braided, clearwater river that flows from the northeast to southwest into the lower Noatak River north of Kotzebue, Alaska. The headwaters drain rocky, alpine tundra terrain of the western Brooks Range. Downstream, the river drains broader valleys with a mixture of boreal spruce forest and tundra vegetation. The watershed is underlain by shallow bedrock and permafrost is generally ice-poor (O'Donnell et al., 2016).

## 2.2 Synoptic Sampling Campaign Design

### 2.2.1 ARC LTER Sites

Our sampling of the TFS watershed networks was designed to capture 30-50 "nested" subcatchments within the Kuparuk River, and Oksrukuyik and Trevor Creeks. Site selection was based primarily on (1) presence of flowing surface waters, (2) representation across varying subcatchment drainage areas, and (3) site accessibility. Often, we *a priori* chose sites located at subcatchment confluences, sampling both upstream locations and then downstream of river mixing. In each of the TFS watersheds, we performed 5 repeated synoptic campaigns, sampling each stream network in August 2016, June 2017, August 2017, June 2018, and August 2018 (exact dates in Table 2). We accessed sampling sites either on-foot or by helicopter within a 6-hour period.

### 2.2.2 NPS/USGS Sites

Sampling of the NPS/USGS watershed networks was designed to capture ~5-10 subcatchments within the Agashashok, Cutler, and Akillik Rivers. Sites were selected to span a gradient of size (subcatchment area, stream order), vegetation (forest



vs. tundra), and permafrost characteristics (parent material, ground ice content). Due to variation in watershed aspect, streams also spanned a spatial gradient in permafrost ground temperatures, areal extent, and active layer thickness (Panda et al., 2016; Sjöberg et al., 2021). In addition to stream chemistry parameters, stream discharge was measured, and samples

were collected to characterize stream biota (benthic biofilm, macroinvertebrates, and resident juvenile fish).

In each of the NPS/USGS watersheds, we performed 4-10 repeated synoptic campaigns, sampling each stream network in June, August, and September 2015; June, August, and September 2016; June and August 2017; and June and August/September 2018 (exact dates in Table 2). We accessed sampling sites by helicopter within a 24- to 96-hour period.

## 3 Methods

### 3.1 Synoptic Site Characterization
### 3.1.1 Subcatchment Delineation for Drainage Area
The location of each stream sampling site was recorded in a spreadsheet and imported into GIS software (ESRI ArcGIS v. 10.4). These sites served as starting points ('pour points') from which watersheds and subcatchments were delineated following the general procedure described here:

(https://support.esri.com/en/technical-article/000012346). The following two digital elevation models (DEMs) were needed to cover the spatial distribution of the stream sampling sites and were used to create the necessary flow direction and flow accumulation layers: ArcticDEM from the Polar Geospatial Center (Porter et al., 2018) and ASTER GDEM v.2 (NASA/METI/AIST/Japan Spacesystems and US/Japan ASTER Science Team, 2009). A Python script was written to iterate over the list of sample sites and execute the watershed delineation procedure.

### 3.1.1 Estimation of terrestrial catchment characteristics for TFS sites
We characterized the terrestrial environment of the TFS sites using remotely sensed data pertaining to the vegetation and topography of each subcatchment. For each subcatchment polygon, we extracted the mean, standard deviation, and range of the elevation, slope, and topographic position index (i.e., the elevation of a given pixel relative to surrounding pixels, sometimes known as slope position). These metrics were calculated from 25-meter-resolution elevation data retrieved from

the USGS National Map website (https://viewer.nationalmap.gov/basic/). The normalized difference vegetation index (NDVI), which indicates the presence of green vegetation, was derived from imagery acquired in summer 2012 by the ETM+ sensor on Landsat 7 (courtesy of the USGS). We also extracted percent cover of vegetation classes in each

subcatchment from the 30-meter-resolution Jorgenson northern Alaska ecosystems map (Muller et al., 2018). All data

extraction was performed using zonal statistics via ArcPy (ESRI, 2016) in Python.

### 3.2 Water Sampling & Analysis

#### 3.2.1 Field sample collection & preparation

##### 3.2.1.1 ARC LTER

During each synoptic campaign, at each site we measured *in-situ* physiochemical variables (this section) and sampled stream

surface water for chemical analysis (section 3.2.2). All physical water samples were "grab" sampled directly from the stream

thalweg, or as close to mid-channel as could be safely accessed. We collected samples in acid-washed and triple-rinsed 1-L

amber PCTE bottles. We used handheld YSI ProPlus multiparameter probes (YSI Instruments Part No: 626281) and YSI

ProODO Dissolved Oxygen Meter (YSI Instruments Part No: 6050020) to measure specific conductance (μS/cm), pH,

temperature (ºC), and dissolved oxygen (DO, in % saturation and mg $O_2$/L) at each sampling site. We placed the probe into

the water column where the water sample was taken and waited for the temperature and DO readings to stabilize before

recording the final value.

Upon returning to the lab at TFS, we processed each water sample into aliquots for specific analytes within 8 hours

of collection. We lab-filtered samples for dissolved water chemistry and nutrients using handheld 60 mL syringes. We triple-

rinsed syringes with unfiltered sample water. Then, we sparged each filter cartridge with ~10 mL of sample water prior to

sample filtration; we used the sparge volume as the initial bottle rinse. We filtered samples for DOC/TDN into triple-rinsed

amber 60-mL HDPE bottles using a 25 mm 0.2 μm cellulose acetate filter (Sartorius CA membrane, 11107-25-N). We

filtered samples for dissolved nutrients, anions, and cations into triple-rinsed clear HDPE 60-mL bottles using a 47 mm 0.7

μm glass fiber filter (Whatman GF/F, 1825-047). Additionally, we placed ~60-mL of unfiltered sample water into a clear

HDPE bottle for analysis of turbidity (NTU) and alkalinity (mg $CaCO_3$/L). After processing, we froze samples at -4 ºC until

analysis, with the exception of aliquots for DOC and total dissolved nitrogen (TDN). We stored DOC/TDN samples at 2 ºC

until analysis. Samples were shipped express to the University of Vermont (UVM) and Brigham Young University (BYU)

for further analysis.

##### 3.2.1.2 NPS/USGS



While sample collection and processing were similar between the TFS and NPS/USGS field sites, the filtration step varied slightly. For NPS/USGS samples, we followed standard USGS protocols. We filtered all samples for nutrient, anion, and

cation analysis using 0.45-µm capsule filters (Geotech Veraspor dispos-a-filter) into 250- or 500-mL HDPE bottles. We filtered samples for DOC and TDN into 125-mL amber glass bottles. Samples for alkalinity and total Fe were left unfiltered. DIC samples were collected without filtering or any headspace in 60-cc luer-lock syringes fit with two-way stopcocks. After processing, we froze samples at -4 ºC until analysis, with the exception of aliquots for DOC, TDN, and DIC that were stored at 2 ºC until analysis. Samples were shipped express to Oregon State University's Cooperative Chemical Analytical

Laboratory (CCAL; http://ccal.oregonstate.edu/) or the USGS in Boulder, Colorado, for further analysis.

### 3.2.2 Dissolved water chemistry analysis

### 3.2.2.1 ARC LTER

We include further detail on analytical methods and instrumentation in Table 3, though we briefly describe our methods here. We measured DOC (as non-purgeable organic carbon, nPOC) and total dissolved nitrogen (TDN) with a total carbon

analyzer (Shimadzu TOC-LCPH with a Total Nitrogen analyzer and ASI-L autosampler). We determined dissolved organic matter (DOM) optical properties including the spectral ratio ($S_r$, unitless) and specific ultraviolet absorbance at 254 nm ($SUVA_{254}$) from the TOC/TN dataset (Helms et al. 2008, Hansen et al. 2016). We colorimetrically analyzed SRP, particulate phosphorous (PP), and total dissolved phosphorous (TDP) on a spectrophotometer (Shimadzu UV-2600). We quantified inorganic nitrogen species (nitrate, $NO_3$; ammonium, $NH_4^+$) using a flow-through injection analysis (Lachat Quikchem Flow

Injection Analysis System). We measured several cations ($Na^+$, $Li^+$, $K^+$, $Mg^{2+}$, $Ca^{2+}$, $NH_4^+$), anions ($F^-$, $Cl^-$), oxoanions ($NO_2^-$, $SO_4^{2-}$, $NO_3^-$, $PO_4^{3-}$) and organic acids (acetate, $CH_3COO^-$; and formate, $HCOO^-$) on an ion chromatrography system (Thermo Fisher Scientific Dionex ICS5000). We quantified other geogenic anions and cations (e.g., $Al^{3+}$, $As^{3-}$, $B^{3-}$, $Ba^{2+}$, $Br^+$, $Ca^{2+}$, $Cd^{2+}$, $Co^{2+}$, $CrO4^-$, Total Cu, Total Fe, $K^+$, $MoO3^{2-}$, $Mg^{2+}$, $Mn^{2+}$, $Na^+$, $Ni^{2+}$, P, $Pb^{2+}$, $S^{2-}$, $Se^{2-}$, $Si^{4+}$, $Sn^{2+}$, $Sr^{2+}$, Ti, V, $Zn^{2+}$) on an ion chromatography inductively coupled plasma mass spectrometer (IC-ICP-MS, iCAP 7000 series, Thermo Scientific).

To estimate turbidity (NTU), we used a Forest Technology Systems (FTS) DTS-12 digital turbidity sensor. We analyzed all samples at room temperature after allowing them to thaw on a lab bench for 2-4 h prior to analysis.

### 3.2.2.2 NPS/USGS

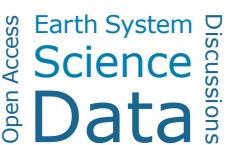

We include further detail on analytical methods and instrumentation in Table 3. For the NPS/USGS sites, we measured DOC

and DIC (O.I Analytical Model 700 TOC Analyzer and Shimadzu TOC-VCSH Combustion Analyzer, respectively). We

characterized DOM aromaticity by measuring UV-visible absorbance on filtered stream water samples on an Agilent Model

8453 photodiode array, and then calculating $SUVA_{254}$ (Weishaar et al., 2003). We also measured TDN and TDP on a

Technicon Auto-Analyzer II. We quantified inorganic nitrogen species ($NO_3^-+NO_2^-$ and unionized $NH_3$) and orthophosphate

($PO_4^{3-}$) using a flow-through injection analysis system (Lachat Quikchem 8500). We calculated alkalinity using a titration to

4.5, using 0.02N $Na_2CO_3$ and 0.02 N $H_2SO_4$ (ManTech PC-Titrate Auto Titrator System). Finally, we used ion

chromatography to measure $Cl^-$ and $SO_4^{2-}$ (Dionex 1500 IC) and absorption spectroscopy to measure $Na^+$, $K^+$, $Mg^{2+}$, $Ca^{2+}$,

and total Fe (Shimadzu AA-7000).

### 3.3 Estimation of secondary ecosystem metrics

In addition to reporting solute concentrations for each synoptic campaign (e.g., Figures 4-5), we estimated secondary

ecosystem metrics for each nested site and watershed. Across these analyses, we assigned any value below detection as the

values of half the limit of quantification and kept these data points in the analysis. When the sample was not run for a

specific solute, the cell was left blank.

### 3.3.1 Subcatchment Leverage

First, we estimated *subcatchment leverage* from each of the synoptic sampling events for each solute. Subcatchment leverage

is calculated as the difference in terms of concentration at each site ($C_s$) from the concentration at the watershed outlet ($C_o$),

subcatchment area ($A_s$) relative to the entire watershed area ($A_o$), and specific discharge at the sampling location ($q = Q_s/A_s$,

where $Q_s$ and $A_s$ are the discharge and subcatchment area at the sampling point):

$$Specific\ Subcatchment\ Leverage = \left[(C_s - C_O) * {}^{A_s}/_{A_O} * q\right] \tag{1}$$

In the case of Eqn. 1, leverage is expressed in units of flux (mass/volume/time). However, if specific discharge is unavailable

for each sampling location leverage can be estimated using only variability in concentration and subcatchment area, so long

as specific discharge ($q$) is similar between subcatchments (Asano et al., 2009; Karlsen et al., 2016). With the exception of

the Agashashok River, which has flow generated from deeper flowpaths, our study watersheds have very little regional

groundwater influence (Lecher, 2017), and the synoptic campaigns were performed near base-flow conditions. Therefore, for



the purposes of this study, we assumed that $q$ was similar for subcatchments within a study watershed, but not necessarily

across the six study watersheds. This assumption was tested at all ARC LTER sites using dilution gauging at a subset of sites

in summer 2018, where we found that values of specific discharge were similar across subcatchment sizes (Shogren,

*unpublished data*). We used Eqn. 2 to estimate subcatchment leverage for all sampling locations across sampling events:

$$Subcatchment\ Leverage = \left[ (C_s - C_O) * {A_s}/{A_O} \right] \qquad (2)$$

Here, subcatchment leverage has units of concentration, or percentage when normalized to outlet concentration. A positive

value for subcatchment leverage is indicative of a net removal along the watershed relative to the concentration at the

watershed outflow, while conversely, a negative value suggests solute production and transport along the watershed (Abbott

et al., 2018). We report both mean leverages for each catchment (presented in Figures 6 and 7) and site-specific

subcatchment leverages for each solute (Figure 10 for DOC and $NO_3^-$, but all other solutes can be found within the secondary

metrics datasets).

**3.3.2 Concentration Variance Collapse**

Next, to assess the representative "patch" size where concentration variance is reduced, we determined the threshold of

concentration *variance collapse* for each solute from each synoptic sampling event (shown in Figure 8). Using

concentrations plotted over watershed area, we used the 'changepoint' package in R (Killick and Eckley, 2014) to determine

the statistical collapse in variance of concentration across the whole watershed area. The variance collapse threshold is

therefore expressed in units of area (here as $km^2$). We used the pruned exact linear time (PELT) method, which compares

differences in data points to determine statistical breakpoints (Abbott et al., 2018; Shogren et al., 2019). We performed this

analysis using scaled concentrations, which were scaled by subtracting the whole watershed mean and dividing by the

standard deviation to facilitate comparison of changes in variance and evaluate convergence towards the watershed mean. A

non-significant variance collapse threshold can be interpreted to mean either the processes controlling lateral fluxes are

operating at too small or too large of a scale to be captured using a subcatchment sampling approach.

**3.3.3 Spatial Stability**

Lastly, we analysed this spatially rich synoptic data to quantify the *spatial stability* of stream nutrient concentrations and to

determine the level of sub-grid resolution necessary to represent controls on lateral nutrient loss. The spatial stability metric



indicates whether spatial sampling is representative or whether spatial patterns "reshuffle" over time. Spatial stability ($r_s$) is

calculated as:

$$(r_s) = \left( \frac{covariance(rg_x, rg_y)}{\sigma_{rg_x} \sigma_{rg_y}} \right) \tag{3}$$

Where $rg_x$ is the rank correlation of subcatchments at the time of synoptic sampling, $rg_y$ is the rank of the long-term flow

weighted concentrations, and s is the standard deviation. We calculated spatial stability using the correlation function in R

(Version 3.3.0), using the Spearman method (Abbott et al., 2018; Shogren et al., 2019). For the purposes of the ARC LTER

analysis, we estimated spatial stability as the Spearman's correlation between Early (June) and Late (August) site

concentrations, resulting in a single spatial stability metric ($r_s$) for 2017 and 2018. For the NPS/USGS sites, spatial stability

was calculated as the correlation between site locations sampled in the Early (June) and Mid (July) and the Mid to Late

(August or September) seasons.

### 3.4 Use and interpretation of secondary ecosystem metrics

The original intent of this manuscript was to present our unique Arctic datasets and showcase the utility of a synoptic

framework in combination with metrics that describe the spatial distribution of river chemistry. To further highlight how

these metrics can inform future sampling design and address fundamental ecological questions, below we describe patterns

for DOC and $NO_3^-$ in the TFS watersheds.

For solutes, the spatial variability in concentration depends on the strength and connectivity of both source and sink

patches superimposed on the structure of the stream network (Abbott et al., 2018). When we plot solute concentration

against subcatchment area, we find more variability water chemistry in smaller subcatchments (<30 km$^2$). This can be

interpreted as a spatial "fingerprint" and is shown most clearly in Figure 10, which displays the spatial distribution of DOC

and $NO_3^-$ concentrations across watersheds and sampling campaigns. Generally high concentration variability in smaller

headwaters, which converges to mean watershed behaviour towards the catchment outlet holds with the conceptualizations

of large rivers as "chemostats" (Creed et al., 2015). In the context of Arctic watersheds, these concentration/area

relationships reveal consistently high DOC and low $NO_3^-$ concentrations in the low-gradient tundra watersheds (Kuparuk

River and Oksrukuyik Creek), despite high variability in smaller contributing subcatchments. In contrast, the alpine



watershed Trevor Creek, has relatively low DOC and high $NO_3^-$ concentrations, likely due to shorter and faster hydrologic flowpaths and lower terrestrial biomass (Shogren et al., 2019). Overall, these findings are consistent with studies that

indicate that slower, longer flowpaths and productive terrestrial vegetation control carbon and nutrient transfer and mobilization in lower-gradient tundra watersheds (Shogren et al., 2019, 2021). If we assume that spatial variability in stream network water chemistry depends primarily on the extent and connectivity of upstream sources/sinks, then the patches sizes that control solute fluxes can be assessed by the spatial scale of the variance collapse (Abbott et al., 2018; Shogren et al., 2019). Across all three TFS watersheds, the generality of variance collapse at intermediate scales is indicative that

subcatchment scale "patches" (~10-50 km$^2$) control whether carbon and inorganic nitrogen is produced or removed at the watershed scale (Figure 10). In addition, the consistency of the thresholds across sampling campaigns (Figure 8 and 10) highlights the importance of capturing intermediate scale biogeochemistry to bridge understandings from plot-level experimentation to larger more regional-scale observations (Shogren et al. 2019).

       When we convert concentrations into estimates of subcatchment leverage (Figure 11), patterns emerge that further

contextualize the spatial distribution of DOC and $NO_3^-$ concentrations. Essentially, we can use subcatchment leverage to reveal the effect of each individual subcatchment contributions on what we "see" at the watershed outlet. This can be interpreted similarly to statistical leverage, where one or more points may exert high influence on a linear regression. Across all TFS watersheds, there are a few select subcatchments that contribute disproportionately to DOC fluxes, while the more variable patterns for $NO_3^-$ suggest additional spatial and seasonal controls (Figure 11). For example, patterns in the Kuparuk

River and Oksrukuyik Creek (Figure 11a-b) could be interpreted to mean that DOC is relatively "leaky" in lower gradient landscapes, while lateral fluxes of $NO_3^-$ are more tightly controlled by biotic demand (Harms et al., 2016; Khosh et al., 2017; Connolly et al., 2018). Across solutes and watersheds, the information gleaned from the leverage metric is useful in several ways. First, subcatchment leverages allow for the direct identification of watershed areas that are disproportionately driving carbon and nutrient exports. For any chosen solute or suite of materials, sites identified as "high leverage" indicate strong

source/sink behaviour, which could be (1) validated with regular field observations that relate riparian or terrestrial conditions with empirical measurements of water chemistry, (2) selected for further study designed to identify the abiotic and biotic mechanisms that drive patterns of riverine chemistry, and/or (3) identified as non-representative sites relative to



proximal subcatchments of similar size and terrestrial characteristics. Relatedly, estimating subcatchment leverage enables researchers to identify sites that are representative of watershed-scale behaviour, which could be used to more effectively

scale biogeochemical dynamics in Arctic rivers relative to outlying subcatchments (Kicklighter et al., 2013). In addition, we can investigate whole watershed behaviour by collapsing the spatial patterns of leverage into a boxplot (as in Figures 6 and 7), with each bar indicating net biogeochemical source/sink patterns for each sampling event. When visualized as "mean" behaviour, the watershed and season-dependent directionality of net leverage patterns are congruent with emerging evidence that landscape template exerts strong control on biogeochemical signals in Arctic rivers (Vonk et al., 2019; Tank et al., 2020;

Shogren et al., 2021).

        Finally, the application of the simple spatial stability metric can help researchers determine whether a sampling location is behaving consistently, or if solute contributions are moving in space across sampling events (Abbott et al., 2018; Dupas et al., 2019). In the context of work in remote watersheds, the ability for researchers to identify both stable and unstable processes presents an exciting opportunity to ask questions about the consistency of subcatchment contributions and

optimize sampling or experimental design. For example, DOC concentrations are generally spatially stable between early and late sampling events ($r_s > 0.50$), particularly in the Kuparuk River and Trevor Creek watersheds (Figure 9). In these landscapes, a high rank correlation indicates that repeated sampling of the same location will result in a similar spatial distribution of concentrations. While sampling repeatedly in the early and late seasons may reveal increases or decreases in solute concentrations (Shogren et al. 2019), the high degree of relatedness indicates that these patterns will be maintained

across the watershed network. However, the general instability ($r_s < 0.50$) for DOC in the Oksrukuyik Creek watersheds signifies substantial spatial shifts across the early and late thaw season (Shogren et al. 2019). While there was variability in the stability/instability across watersheds and solutes, the stability metric can be used by future researchers to identify whether sapling the same location repeatedly does or does not represent the spatial dynamics across sampling events.

## 4. Data Availability

The data from the NPS/USGS are available at https://doi.org/10.5066/P9SBK2DZ (O'Donnell et al., 2021). Data from TFS are stored at the Environmental Data Center data repository (http://dx.doi.org/10.6073/pasta/258a44fb9055163dd4dd4371b9dce945) (Abbott et al., 2021).

## 5. Conclusions

With this work, we provide a detailed characterization of physical, chemical, and biological parameters that are essential to

the study of river solute production and removal. Further, we derive novel metrics from these data that describe the spatio-

temporal patterns of watershed biogeochemistry in six permafrost-underlain Arctic watersheds. These data represent one of

the most extensive river chemistry datasets from understudied permafrost-dominated regions as well as a state-of-the-science

watershed characterization conducted at unprecedented spatial resolution. Taken together, these data will enable the

generation of novel hypotheses and model assessments for watershed/landscape relationships from diverse Arctic watersheds

in future studies.

**Author contributions:** All co-authors participated in the field collection, laboratory analysis, and/or curation of the dataset,

and contributed to writing and/or editing this paper. AJS was primarily responsible for writing this paper and assembly of the

archival database.


**Competing Interests:** The authors declare that they have no conflict of interest.

**Acknowledgements:** Data and facilities were provided by the ARC LTER at TFS, NPS, and USFS. The authors gratefully

acknowledge the ARC LTER, TFS staff, and CH2M HILL Polar Services for assistance in support of this work. The authors

also thank the National Park Service's Western Arctic Parklands in Kotzebue, Alaska, for assisting with logistics and

permitting. DEMs were provided by the Polar Geospatial Center under NSF-OPP awards 1043681, 1559691, and 1542736.

**Financial Support:** The authors would also like to acknowledge their funding support: AJS (NSF-DBI-1906381, NSF-OPP-

1916567); JPZ and TW (NSF-EAR-1846855, NSF-OPP-1916567); BWA (NSF-OPP-1916565), WBB (NSF-OPP-1916576,

NSF DEB-1637459); and FI and AM (NSF DEB-1637459). This work was supported by the Changing Arctic Ecosystems

Initiative of the Wildlife program of the USGS Ecosystems Mission Area. Additional support was provided by the NPS



Arctic Inventory & Monitoring program. Any use of trade, firm, or product names is for descriptive purposes only and does

not imply endorsement by the U.S. Government.




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



## Figures & Tables

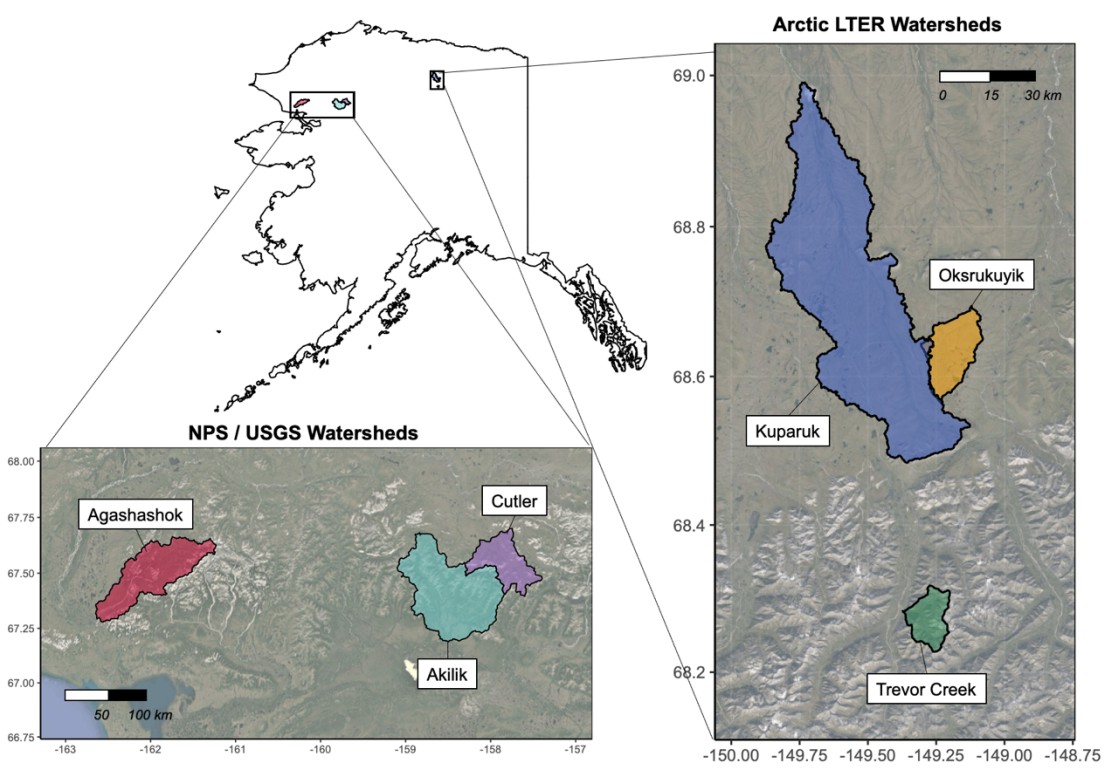


**Figure 1. Regions of northern Alaska associated with the Arctic Long-Term Ecological Research (ARC LTER) site at Toolik Field Station (TFS) and National Park Service (NPS) and U.S. Geological Survey (USGS) watersheds. Map created in R Studio (version 1.2.1335) with base imagery from ESRI and © Google Earth (Version 7.3.3.7786).**

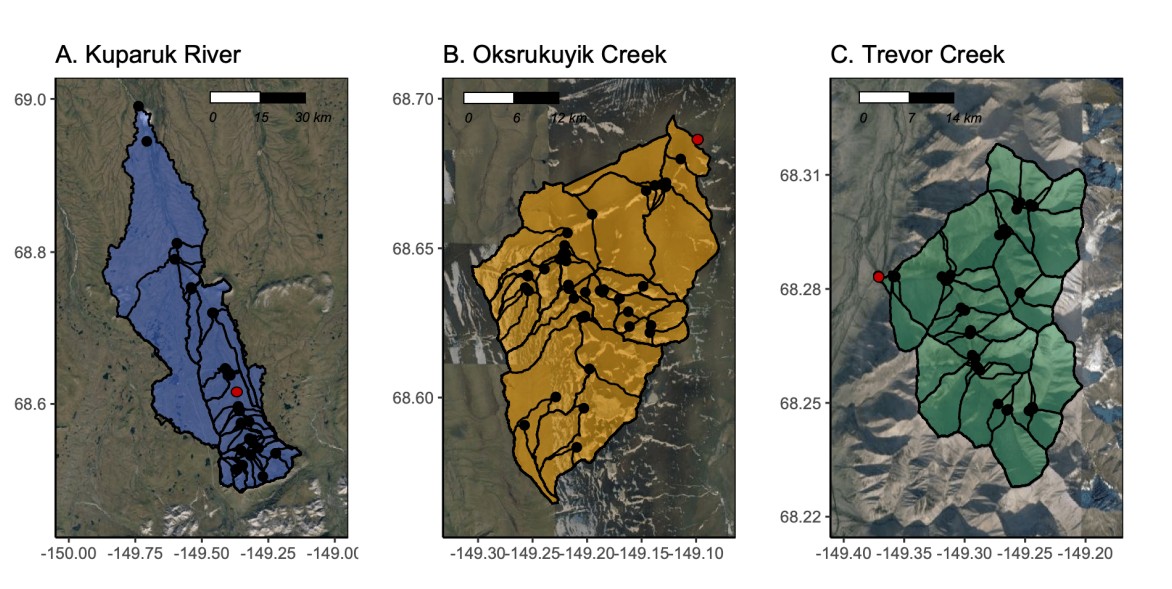

**Figure 2. Synoptic sampling sites (black points) with subcatchment delineations from three watersheds related to the Arctic Long-Term Ecological Research (ARC LTER) site at Toolik Field Station (TFS) on the North Slope of Alaska. Study watersheds include the A. Kuparuk River (blue), B. Oksrukuyik Creek (orange), and C. Trevor Creek (green). Scale bars in km. The ARC LTER monitoring stations are denoted by red points and described further in Shogren et al. 2021. Map created in R Studio (version 1.2.1335) with base imagery from ESRI and © Google Earth (Version 7.3.3.7786).**



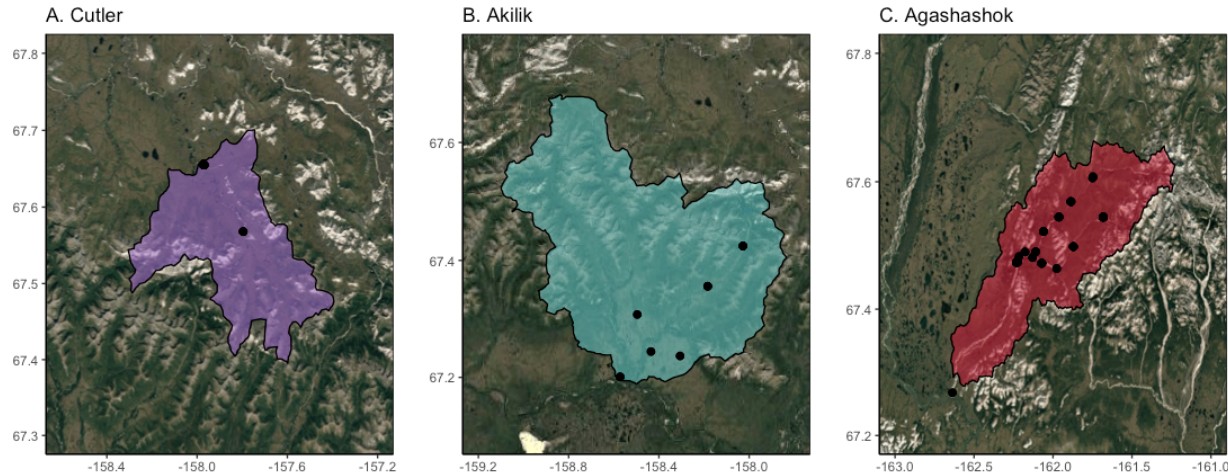

**Figure 3. Synoptic sampling sites in three NPS/USGS watersheds. Study watersheds include the A. Cutler, B. Akilik,**
**and C. Agashashok Rivers. Map created in R Studio (version 1.2.1335) with base imagery from ESRI and © Google**
**Earth (Version 7.3.3.7786).**
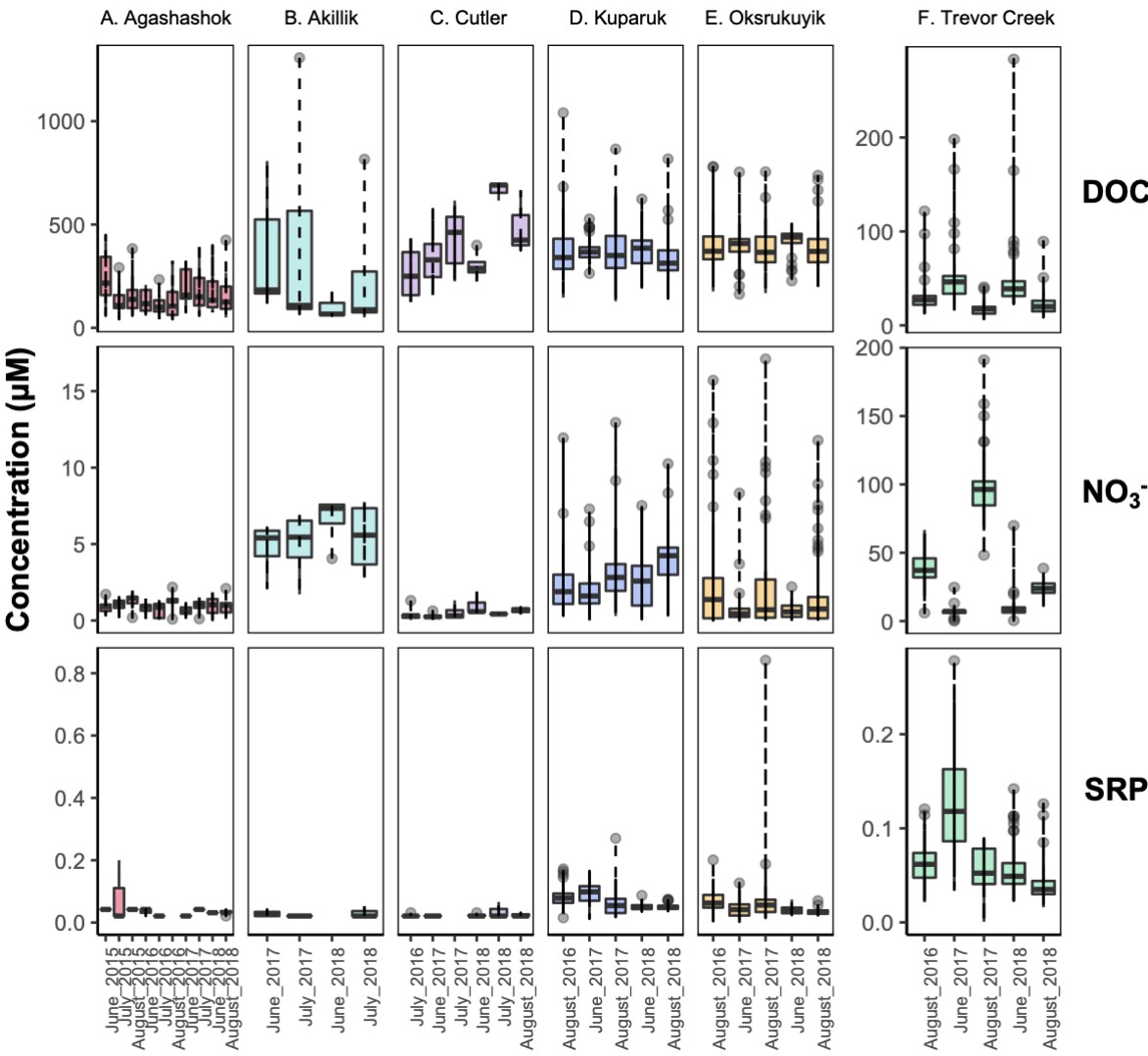

**Figure 4: Boxplots of dissolved organic carbon (DOC, top row), nitrate (NO$_3^-$, middle row), and soluble reactive phosphorus (SRP, bottom row) concentration ranges (in µM) in the (A) Agashashok River, (B) Akilik River, (C) Cutler River, (D) Kuparuk, (E) Oksrukuyik Creek, and (F) Trevor Creek watersheds across all years and seasons sampled.**

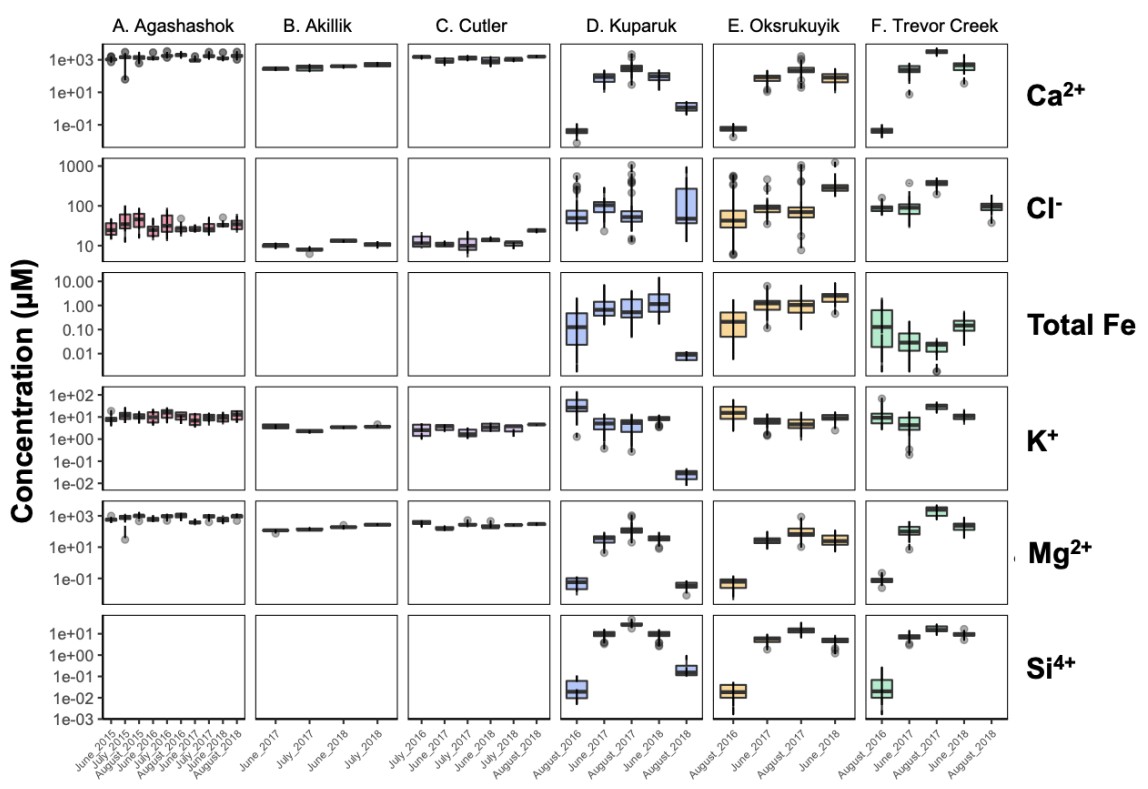


**Figure 5: Boxplots of Ca²⁺, Cl⁻, Total Fe, K⁺, Mg²⁺, and Si⁴⁺ concentration ranges (in μM) in the (A) Agashashok River, (B) Akilik River, (C) Cutler River, (D) Kuparuk River, (E) Oksrukuyik Creek, and (F) Trevor Creek watersheds across all years and seasons sampled.**



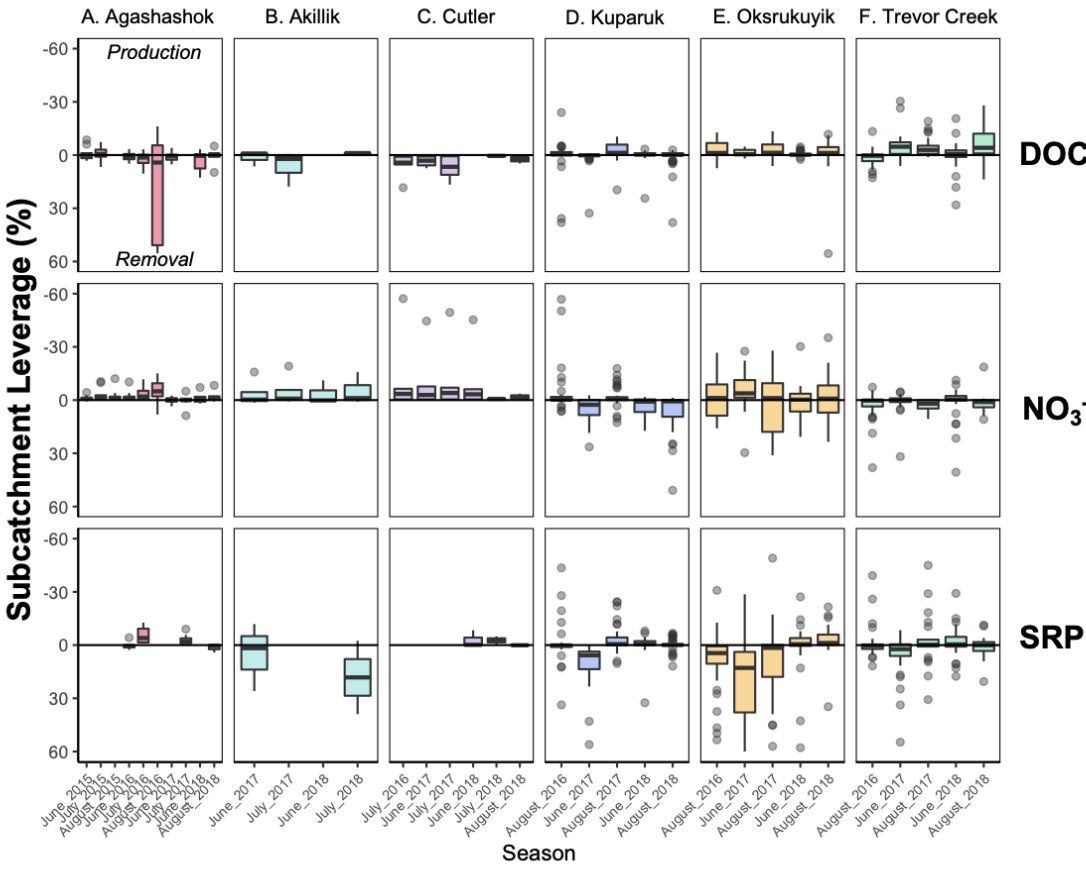

**Figure 6: Boxplot of subcatchment leverage for select reactive solutes (DOC, NO₃⁺, and SRP) in the (A) Agashashok River, (B) Akilik River, (C) Cutler River, (D) Kuparuk River, (E) Oksrukuyik Creek, and (F) Trevor Creek watersheds across all years and seasons sampled. Note reversed axes for ease of interpretation: negative values above the 0 line indicate production, positive values below the 0 line indicate removal.**

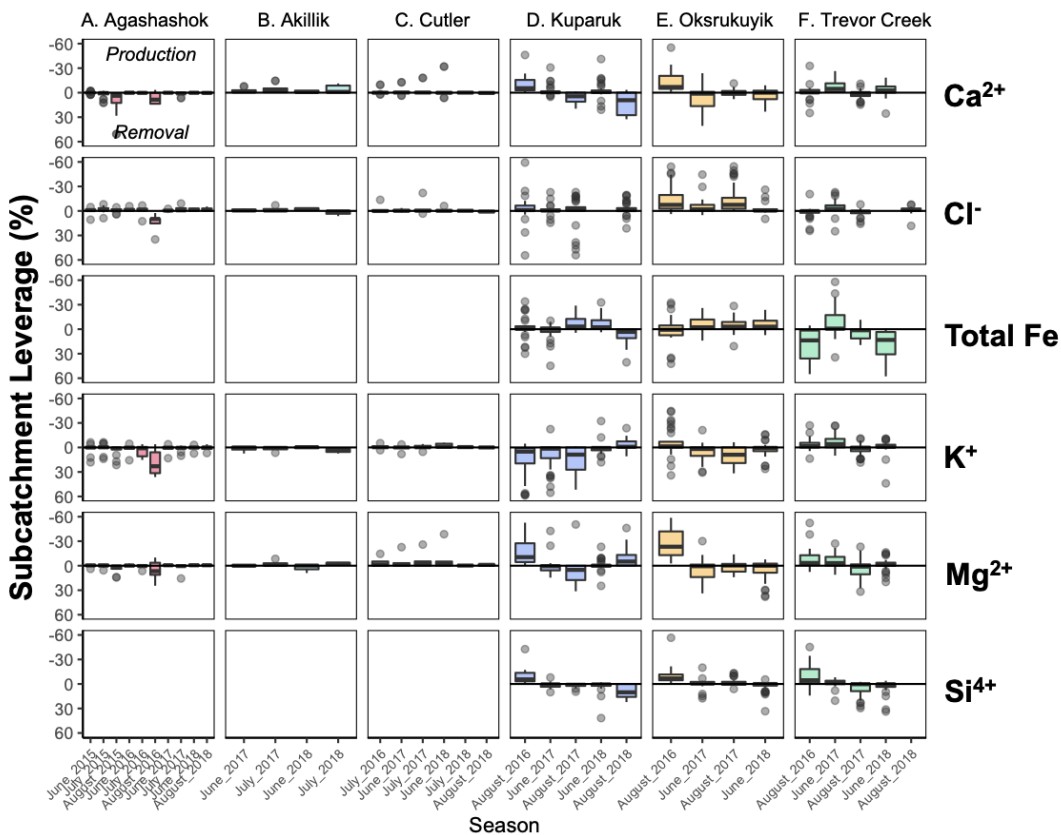

**Figure 7: Boxplot of subcatchment leverage for select conservative solutes ($Ca^{2+}$, $Cl^-$, Total Fe, $K^+$, $Mg^{2+}$, and $Si^{4+}$) in the (A) Agashashok River, (B) Akilik River, (C) Cutler River, (D) Kuparuk River, (E) Oksrukuyik Creek, and (F) Trevor Creek watersheds across all years and seasons sampled. Note reversed axes for ease of interpretation: negative values above the 0 line indicate production, positive values below the 0 line indicate removal.**



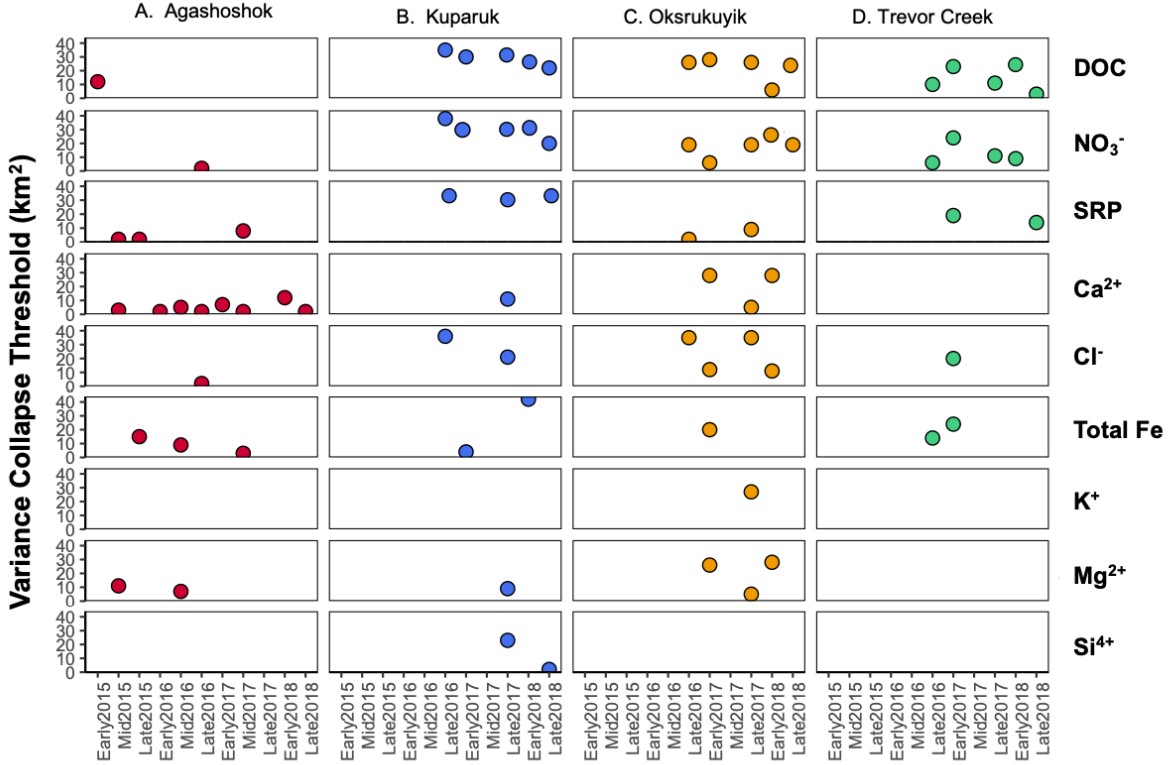

**Figure 8: Figure 8: Scatter plot of variance collapse threshold for each repeated sampling for the A. Agashashok River, B. Kuparuk River, C. Oksrukuyik Creek, and D. Trevor Creek watersheds for select reactive (e.g., DOC, $NO_3^-$, and SRP) and conservative solutes ($Ca^{2+}$, $Cl^-$, Total Fe, $K^+$, $Mg^{2+}$, and $Si^{4+}$). When data were not present, there was no significant collapse detected. Variance collapse thresholds are not shown for the Akilik and Cutler Rivers, as these thresholds were often non-significant.**





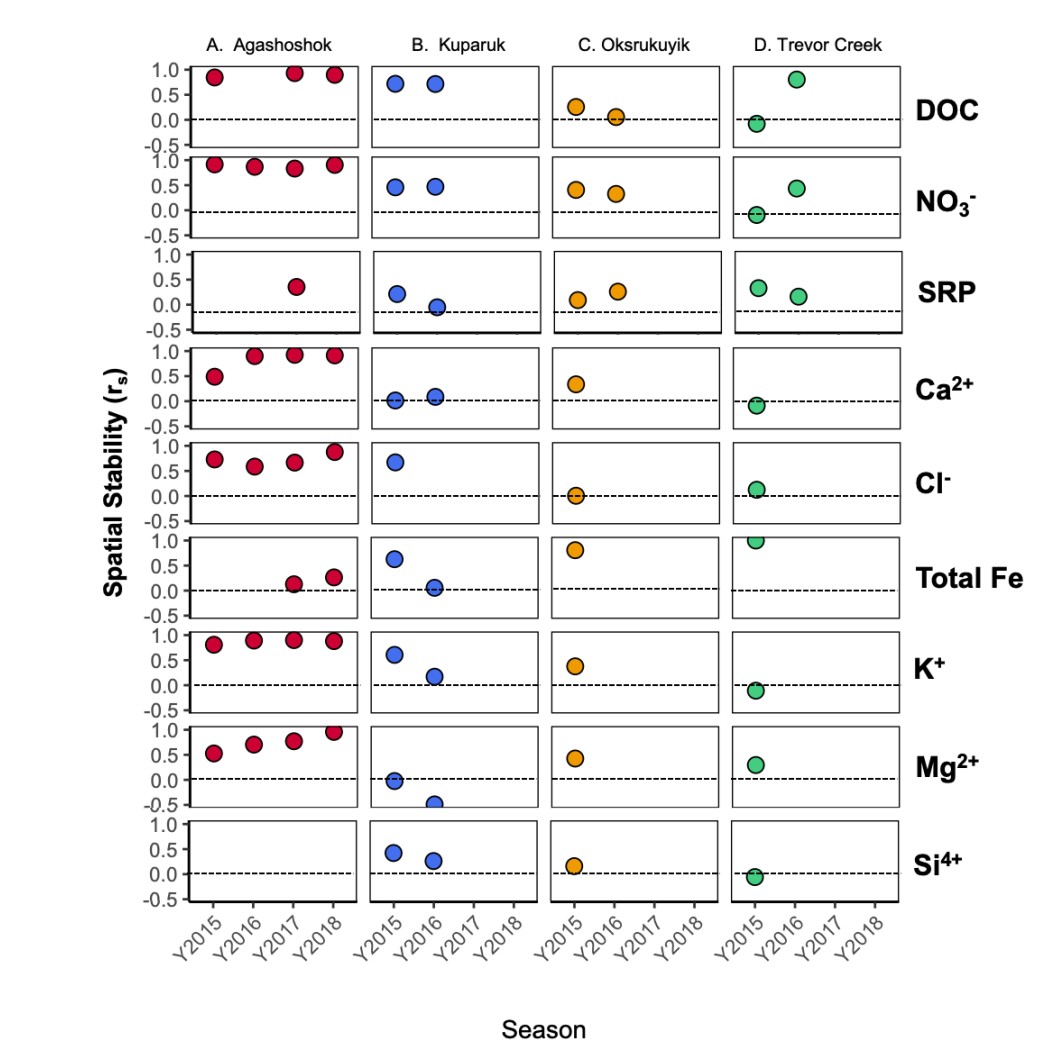

**Figure 9: Scatter plot of spatial stability ($r_s$) for each repeated sampling for the A. Kuparuk River, B. Oksrukuyik Creek, and C. Trevor Creek watersheds for select reactive (e.g., DOC, $NO_3^-$, and SRP) and conservative solutes ($Ca^{2+}$, $Cl^-$, Total Fe, $K^+$, $Mg^{2+}$, and $Si^{4+}$). When data were not present, there is no spatial stability reported.**



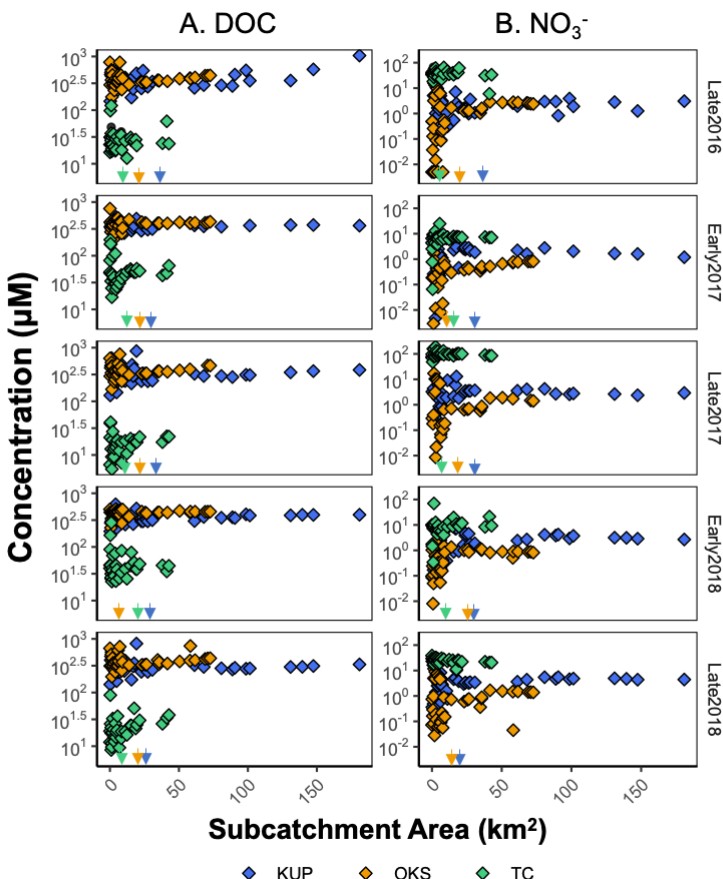

**Figure 10: Scatter plot of log-scale A. DOC and B. NO$_3^-$ concentrations (µM) across subcatchment area (km²) or each repeated sampling in the Kuparuk River (blue points), Oksrukuyik Creek (orange points), and Trevor Creek (green points) watersheds. Significant variance collapse thresholds are represented by a colored arrow.**




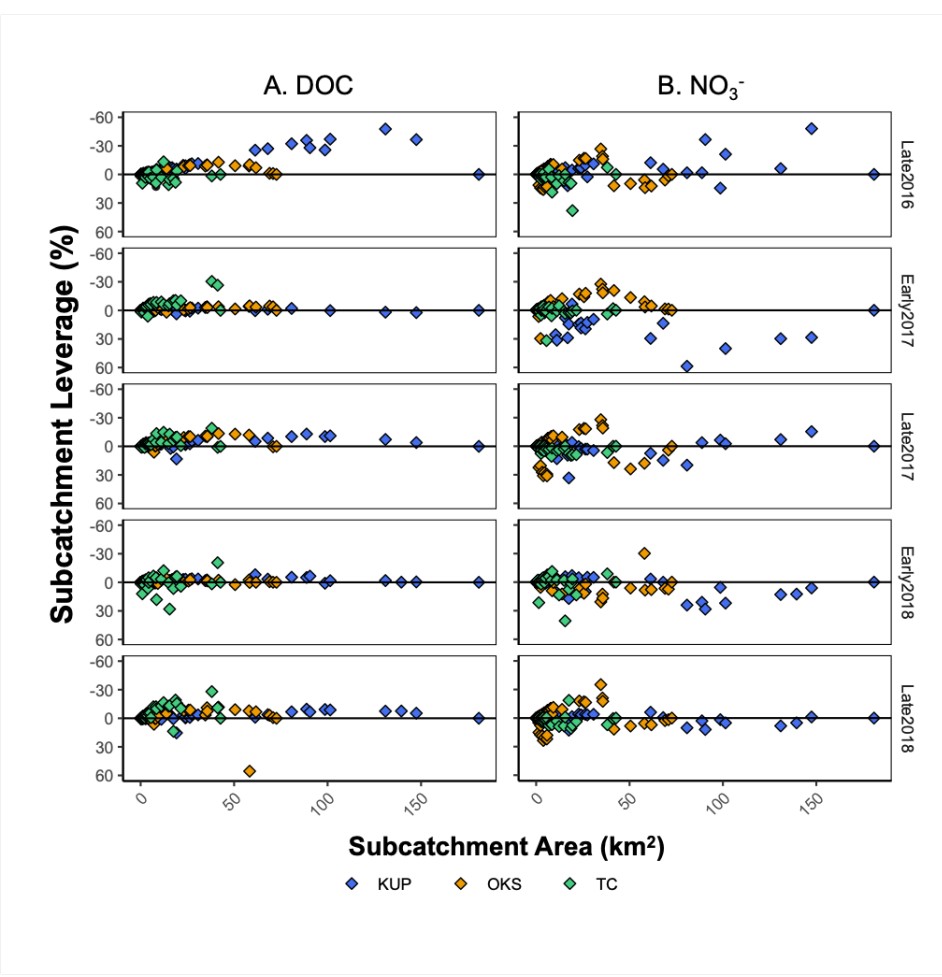

**Figure 11: Scatter plot of A. DOC and B. NO₃⁻ leverages across subcatchment area (km²) or each repeated sampling in the Kuparuk River (blue points), Oksrukuyik Creek (orange points), and Trevor Creek (green points) watersheds. Note reversed axes for ease of interpretation: negative values above the 0 line indicate production, positive values below the 0 line indicate removal.**





**Tables**

**Table 1. Summary of site characteristics for the watersheds where synoptic samplings were conducted. The descriptions are considered representative of the major landform types within the TFS and NPS/USGS watersheds.**

| | Site | Slope (º) | Mean Elevation (m) | Geologic Setting | Permafrost Zone | Primary vegetation | Number of sampling sites | Total Drainage Area (km$^2$) |
|---|---|---|---|---|---|---|---|---|
| TFS | Kuparuk River | Low (3.1) | 988 | Sagavanirktok Old Glaciated Uplands | Continuous permafrost | Wet acidic tundra | 45 | 92.5 |
| | Oksrukuyik Creek | Low (3.2) | 862 | Sagavanirktok Young Glaciated Valleys | Continuous permafrost | Wet acidic tundra | 42 | 72.6 |
| | Trevor Creek | High (9.4) | 1595 | Sagavanirktok Young Glaciated Valleys | Continuous permafrost | Alpine valley | 35 | 42.7 |
| NPS/ USGS | Agashashok River | High (9.3) | 317 | Sedimentary carbonate and non-carbonate lithology | Continuous permafrost | Boreal spruce forest, arctic tundra | 9 | 1058.0 |
| | Cutler River | High (8.0) | 644 | Quaternary, noncarbonate deposits (glaciolacustrine) | Continuous permafrost | Boreal spruce forest, arctic tundra | 6 | 566.7 |
| | Akillik River | High (14.8) | 447 | Quaternary, silt and peat | Discontinuous permafrost | Boreal spruce forest, arctic tundra | 5 | 262.1 |




**Table 2: Description of the sampling campaign regimes, including dates for each campaign, for the TFS and NPS/USGS watersheds.**


| | Site | Years of repeated synoptic sampling | Number of sampling events | Sampling Dates | Seasonal Sampling |
|---|---|---|---|---|---|
| TFS | Kuparuk River | 2016-2018 | 5 | **2016:** 8/26<br>**2017**: 6/5; 8/27<br>**2018**: 6/6; 8/24 | June, August |
| | Oksrukuyik Creek | 2016-2018 | 5 | **2016:** 8/17<br>**2017**: 6/3; 8/24<br>**2018**: 6/4; 8/23 | June, August |
| | Trevor Creek | 2016-2018 | 5 | **2016:** 8/22<br>**2017**: 6/7; 8/31<br>**2018**: 6/8; 8/28 | June, August |
| NPS/ USGS | Agashashok River | 2015-2019 | 10 | **2015:** 6/9-6/12; 8/7-8/11; 9/16-9/19<br>**2016:** 6/7-6/12; 8/9-8/12; 9/8-9/9<br>**2017**: 6/6-6/8; 8/16-8/18<br>**2018**: 6/11-6/12; 9/2-9/6 | June, August, Sept |
| | Cutler River | 2015-2019 | 5 | **2016**: 8/14-8/15<br>**2017**: 6/10; 8/20-8/21<br>**2018**: 6/14; 8/31-9/1 | June, August, Sept |
| | Akillik River | 2015-2019 | 4 | **2017:** 6/11-6/12; 8/22-8/23<br>**2018:** 6/13; 8/30 | June, August, Sept |



**Table 3. Summary of sample processing and analytical methods used for the dataset for A. TFS and B. NPS/USGS field sites. Expanded table in supplement.**


| A. TFS | | | | |
|---|---|---|---|---|
| | **Parameter** | **Units** | **Instrument** | **Analytical Method** |
| **Watershed Characteristics** | Drainage Area | km2 | Arc-GIS | Spatial analysis |
| | MeanSlope | Degrees | | |
| | STDSlope | Degrees | | |
| | MeanRough | NDVI | | |
| | STDRough | NDVI | | |
| **Water Quality Measurements** | Temperature | C | YSI Pro Plus Multiparameter Mete | Analyzed in the field with a handheld field probe |
| | Specific Conductivity | uS/cm | | |
| | pH | pH | | |
| | O2 | % Sat | YSI ProODO Dissolved Oxygen Meter | |
| **Water Chemistry** | Turbidity | NTU | Forest Technology Systems (FTS) DTS-12 digital turbidity sensor | Nephelometric geometry |
| | DOC | uM | | Combustion catalytic oxidation method |
| | TDN | uM | Shimadzu TOC-LCPH with TN | High-Temperature Catalytic Combustion and Chemiluminescence Detectio |
| | N-NO3 | uM | | Cadmium Reduction |



| | | | |
|---|---|---|---|
| N-NH4 | uM | Lachat Quikchem Flow Injection Analysis System | Sodium salicylate-based procedure that requires a standard heating unit and is read at 660 nm; |
| SRP | uM | Shimadzu UV-2600 spectrophotometer | Colorimetric analysis using an ammonium molybdate-based reagent. (Although technically USGS refers to this as the Ascorbic Acid method...) |
| PP | uM | | Same as above, preceded by combustion at 500C and a hydrochloric acid digestion |
| TDP | uM | | Same as SRP, preceded by a potassium persulfate digestion. |
| F, Acetate, Formate, Cl, NO2, Br, SO4, PO4, Li, Na, NH4, K, Mg, Ca | uM | Thermoscientific Dionex ICS-2100 Integrated IC System with Electrolytic Eluent Generation with an AS-AP Autosampler | Ion Chromatography |
| nPOC | mg/L | Shimadzu TOC-LCPH with TN | Combustion catalytic oxidation method |
| Spectral Slope Ratio | Unitless | | |
| SUVA 254 | Absorbance (A) | | |





| Al, As, B, Ba, C, Cd, Co, Cr, Cu, Fe, K, Mg, Mn, Mo, Na, Ni, P, Pb, S, Se, Si, Sr, Ti, V, Zn | uM | iCAP 7000 series, Thermo Scientific | Inductively Couple Plasma (ICP) |
|---|---|---|---|
| Alkalinity | meq/L | Accumet AB200 pH meter | Samples individually titrated with 0.18N sulfuric acid |

**B. NPS**

| | Parameter | Units | Instrument | Specific Method |
|---|---|---|---|---|
| **Watershed Characteristics** | Drainage Area | km2 | Arc-GIS | Watershed delineation |
| | MeanSlope | Degrees | | |
| | STDSlope | Degrees | | |
| | MeanRough | NDVI | | |
| | STDRough | NDVI | | |
| **Water Quality Measurements** | Temperature | C | YSI Pro Plus Multiparameter Mete | |
| | Specific Conductivity | uS/cm | | |
| | pH | pH | | |
| | O2 | % Sat | YSI ProODO Dissolved Oxygen Meter | |
| **Water Chemistry** | DOC | uM | O.I Analytical Model 700 TOC Analyzer | Platinum-catalyzed persulfate wet oxidation method |
| | TDN | uM | Technicon Auto-Analyzer II | Persulfate digest |
| | NO3+NO2 | uM | Labchat QuikChem 8500 | Cadmium reduction |
| | NH3 | uM | | Colorimetric |
| | SRP | uM | | Ascorbic acid method |



| | | | |
|---|---|---|---|
| TDP | uM | Technicon Auto-Analyzer II | Persulfate digest |
| Cl, SO4 | uM | Dionex 1500 IC | ion chromatography |
| Na, K, Mg, Ca, Fe | uM | Shimadzu AA-7000 | flame atomic absorption spectroscopy |
| SUVA 254 | L mgC-1 m-1 | | spectrophotometer |
| Alkalinity | mgCaCo3/L | ManTech PC-Titrate Auto Titrator System | Titration to 4.5, use 0.02N Na2Co3 and 0.02 N H2SO4 |
| DIC | mg/L | Shimadzu TOC-VCSH Combustion Analyzer | |