# Peer review of "Multi-year, spatially extensive, watershed scale synoptic stream chemistry and water quality conditions for six permafrost-underlain Arctic watersheds"

_Earth System Science Data, 2021_

## Author Response (AR1)

**Reviewer 1 Comments**

**Summary**

**The work by Shogren et al. represents a thorough evaluation of a unique (and challenging to collect) dataset using really interesting metrics to compare diverse sites. Specifically, they evaluate synoptic water chemistry across six watersheds in northern Alaska using secondary ecosystem metrics. As I was not familiar with these metrics, I appreciated the descriptions that were provided. The article is a nice introduction to these metrics and presents examples of their application without going into extensive interpretation of each solute, which seems appropriate for this journal. I have provided comments below with respect to the datasets as well as interpretation.**

We thank the reviewer for their careful consideration, as their comments have helped us improve the clarity and quality of our manuscript. Please see our responses to each comment below. All changes noted are included in the revised manuscript are highlighted in yellow.

**Specific comments**

**Regarding the datasets, both datasets are accessible and are well structured and supported by metadata. However, I did not see any indication of detection limits in the Abbott/TFS dataset or data flags that would indicate values below quantification. This omission limits proper use of the dataset (my apologies if I've missed it somewhere?).**

We thank the reviewer for their review of the datasets, and we apologize for the omission. For the Abbott/TFS dataset, we elected to keep values that were at or below detection in the dataset, as these sites were still sampled, and we did not want to introduce a zero or omission bias as low concentration values are still meaningful. Instead, we used values that were half the limit of detection (as noted in Section 3.3). However, we recognize that for others to reliably use the dataset, it should be clear when samples were not taken (missing data are included as -7777 in the datasets) vs when concentrations were very low. We have added the detection limits for each constituent as a column in Table 3, which allow clearer identification of values that are at or below detection. Further, we plan to resubmit a correction to the dataset so that below detection values are more clearly distinguished.

**An underlying assumption of the secondary ecosystem metrics is that measurements at the outlet conservatively integrate measurements of the subcatchments. Given that these solutes may not behave conservatively in the streams, can you speak to how in-stream processes might affect the metrics being used here?**

The reviewer is correct that the subcatchment leverage estimation is effectively normalized to the pre-determined catchment outlet, such that the outlet leverage is effectively 0. However, the metric does not assume that the integration of the metrics is conservative; rather, the metric itself represents the spatially distributed mass balance for each element relative to the outflow, which allows the estimation (though not direct identification) of net biological/physical processes that retain or produce materials in terms of their net export as they are transported and transformed

towards the watershed outlet. In other words, while the values of leverage are dynamic (i.e., they can change depending on the placement of the watershed outlet) and do not identify removal/production mechanisms, the values do indicate whether material is removed or produced at the catchment scale. We have made this clearer in the main text:

First, we can investigate whole watershed ("net") behaviour by calculating the mean leverage and examining the distribution of values with boxplots (as in Figures 6 and 7). As a more specific example, mean $NO_3^-$ leverage within the Kuparuk watershed (Figure 6D, second row) were consistently above zero (note the reversed axis), revealing strong removal or retention before it reached the watershed outlet, which is consistent with high biotic N demand. Within this same watershed, DOC leverage values were often at or just above the zero line (Figure 6D, first row), representing primarily conservative transport of DOC (i.e., no net production or uptake). Within the lake-influenced Oksrukuyik watershed, $NO_3^-$ leverage values were more variable (i.e., leverage above/below zero-line; Figure 6E, second row), implying a combination of removal and production mechanisms acting across the watershed network. When visualized as "net" behaviour, the watershed and season-dependent directionality of net leverage patterns are congruent with emerging evidence that landscape template exerts strong control on biogeochemical signals in Arctic rivers (Vonk et al., 2019; Tank et al., 2020; Shogren et al., 2021). As a compliment to the first approach, we can additionally examine individual subcatchment leverage values to reveal the effect of each contribution on what we observe at the watershed outlet. This can be interpreted similarly to statistical leverage, where one or more points may exert high influence on a linear regression.

To add further clarification, it is still an open scientific question of how subcatchment leverage translates to instream/terrestrial/combined processes; currently, leverage values simply indicate "net" material production/removal, where we are not yet able to clearly link biological and/or physical mechanisms to the direction or value of the leverage estimates. We note that this is work that we are actively pursuing within these watersheds and hope to have further clarity on this point in the coming years.

**It is important to distinguish between nominally dissolved Fe (< 0.7 um) from the TFS sites and total Fe (unfiltered) from the NPS/USGS sites. Is there any information on how these values compare?**

We thank the reviewer for this point. While we are not able to compare values within or across the TFS and NPS/USGS watersheds, we will address this comment by distinguishing the Fe values in the figure captions (Figs. 5, 7, 8, and 9), and throughout the main text.

**I'm unclear on the interpretation of subcatchment leverage. It is described as how subcatchments produce or remove solutes relative to what is measured at the outlet. As such, should the average across subcatchments (e.g., Figures 6 and 7) equal zero, where production and removal are balanced? If these values are not zero, does that indicate that the watershed was not fully captured?**

This is an excellent question. The subcatchment leverage approach essentially represents a distributed mass-balance, which can be used to (1) identify subcatchments exerting strong influence on material fluxes and/or (2) depict net patterns of the entire network. In the case of "net" ecosystem behavior (Figs. 6 and 7), the later interpretation applies. When values are not equal to zero in these figures, in general, many subcatchments are removing/retaining or producing/exporting material relative to the concentrations observed at the watershed outlet. When presented as a bar graph (Figure 7), this can help depict the general direction of solute behavior within a watershed. As a more specific example, in Figure 6D, leverage values for nitrate within the Kuparuk watershed show a bar that is generally far under the zero line, suggesting strong removal of these solutes before they reach the catchment outlet, potentially indicative of strong biotic demand. In this same watershed, leverage values for DOC are at or slightly above the zero line, representing a uniform "leaky" behavior consistent with landscape availability in tundra landscapes with high organic matter.

To clarify this point, we have added the following text to Section 3.4: First, we can investigate whole watershed ("net") behaviour by calculating the mean leverage and examining the distribution of values with boxplots (as in Figures 6 and 7). As a more specific example, mean $NO_3^-$ leverage within the Kuparuk watershed (Figure 6D, second row) were consistently above zero (note the reversed axis), revealing strong removal or retention before it reached the watershed outlet, which is consistent with high biotic N demand. Within this same watershed, DOC leverage values were often at or just above the zero line (Figure 6D, first row), representing primarily conservative transport of DOC (i.e., no net production or uptake). Within the lake-influenced Oksrukuyik watershed, $NO_3^-$ leverage values were more variable (i.e., leverage above/below zero-line; Figure 6E, second row), implying a combination of removal and production mechanisms acting across the watershed network. When visualized as "net" behaviour, the watershed and season-dependent directionality of net leverage patterns are congruent with emerging evidence that landscape template exerts strong control on biogeochemical signals in Arctic rivers (Vonk et al., 2019; Tank et al., 2020; Shogren et al., 2021). As a compliment to the first approach, we can additionally examine individual subcatchment leverage values to reveal the effect of each contribution on what we observe at the watershed outlet. This can be interpreted similarly to statistical leverage, where one or more points may exert high influence on a linear regression.

**I think my question is partially addressed by lines 363 – 365, but I could use some interpretation of that statement. That is, what does it mean in practice for landscape template to exert strong control on biogeochemical signals in Arctic rivers?**

Please see response to the previous comment.

**Using spatial stability as a metric to generate hypotheses about processes driving stability or instability is really intriguing!**

We greatly appreciate the reviewer's enthusiasm about the spatial stability metric. We are similarly excited by the ability to assess spatial patterns.

**Technical comments**

**Line 299. Please quantitively define the "statistical collapse in variance of concentration". As I understand it, this was calculated as the difference between a subcatchment concentration, and the catchment mean divided by the catchment standard deviation. Would a collapse in variance indicate the catchment area at which the stream concentration is within the standard deviation of the catchment mean?**

We thank the reviewer for this comment. We refer to "statistical collapse" as the "breakpoint" or reduction in variance that is determined using a changepoint analysis in R using a Pruned Exact Linear Time (PELT) method. The PELT method detects changepoints by systematically minimizing a "cost" function over iterations of collapse locations (Kilick et al. 2011) and is commonly used to analyze changes in variance structure over time and space. We have added this citation to the text, and clarified the text as: ==Using concentrations plotted over watershed area, we used the 'changepoint' package in R (Killick and Eckley, 2014) to determine the collapse in variance of concentration across the whole watershed area. To determine the reduction in variance statistically, we used the pruned exact linear time (PELT) method, which compares differences in data points to determine statistical breakpoints (Abbott et al., 2018; Shogren et al., 2019).==

**Line 313. Should this be that sigma (not s) indicates standard deviation?**

This is now corrected

**Line 378. Says "sapling" instead of "sampling"**

Fixed

**Equation 1-2. Leverage is said to be expressed in terms of mass/volume/time, but doesn't volume cancel out of equation 1? E.g., mass/volume * area/area * (volume/time)/area = mass/area/time. Since % is used here, that specific equation should be used for equation 2 or placed as equation 3 (e.g., subcatchment leverage (%) = 100*[(C_s − C_o) * (A_s/A_o)]/C_o).**

We thank the reviewer for finding this mistake! We have fixed Equation 2 as suggested.

**Figure 4. Please define the boxplot metrics (for this and all figures). Why are the scales for Trevor Creek different? Ranges for DOC and SRP are comparable to ranges for the other watersheds.**

We have detailed the boxplot metrics for Figures 4, 5, 6, and 7.

We put the Trevor Creek data on different y-axes scales to better show the variability of concentrations within that watershed that was challenging to visualize on the same data ranges as the other watersheds. Trevor Creek has some of the lowest DOC and SRP and the greatest NO3 concentrations. We have chosen to keep the figure as-is.

**Figure 5. Dissolved Si is present as an oxyanion in solution, not a cation. I suggest dissolved Si (or total Si) rather than $Si^{4+}$ here and elsewhere.**

Done

**Please also make y-axes consistent (e.g., 0.1, 1, 10, 100, 1000 or $10^{-2}$, $10^{-1}$, $10^0$, $10^1$, $10^2$, $10^3$) rather than 1e+01 or such.**

Done

**Figure 6. $NO_3^+$ in legend should be $NO_3^-$.**

Fixed

**Figure 8. Different date specifications are used here than elsewhere (month, season, year). Were data partitioned differently?**

Fixed

**"Season" is used as the x-axis label for graphs where it should be "month" or "year".**

Fixed

**Table 3. Needs to be edited for typos (e.g., u instead of mu symbols, superscripts, missing letters, degree symbol missing, units for SUVA)**

We have gone through the table and fixed many typos, units, and sub/superscripts in the table.

**Reviewer 2 Comments**

**Summary:**

**Shogren et al. represent a valuable dataset consisting of water chemistry across six watersheds in northern Alaska. The authors have already addressed all previous reviewer's concerns.**

We thank the reviewer for their careful consideration. Please see our response to your comment below.

**Here, I have only one comment regarding the measurement of "the spatial stability" for your consideration: As shown in Eq.3, Spearman's rho was used to assess the correlation between rgx and rgy, and ggx is the rank correlation of sub-catchments? Is this correct? Moreover, the significance test should be given for Figure 9.**

Yes, the reviewer is correct. We have amended the text as follows: Where $rg_x$ is the rank of subcatchments at the time of synoptic sampling, $rg_y$ is the rank of the long-term flow weighted concentrations, while $\sigma_{rgx}$ and $\sigma_{rgy}$ are the standard deviations of the rank variables.

Further, we have added indication of $r_s$ significance in a revised Figure 9, as suggested.

---

## Author Response (AR2)

**Response to Reviewer / Editor Comments:**

**For the next revision, I kindly ask you to adjust the copyright statement of figures 1-3 as follows: "© Google Earth".**

We have added this change to the manuscript captions for Figures 1-3.